# Upstream interventions to promote oral health and reduce socioeconomic oral health inequalities: a scoping review protocol

Eleanor R Dawson ![ORCID],[1] Michelle Stennett ![ORCID],[1] Blánaid Daly ![ORCID],[2] Lorna M D Macpherson ![ORCID],[3] Paul Cannon ![ORCID],[4] Richard G Watt ![ORCID] [1]

[1]Department of Epidemiology and Public Health, University College London, London, UK
[2]Division of Public and Child Dental Health, Dublin Dental University Hospital and Dental School, Trinity College Dublin, Dublin, Ireland
[3]Department of Dental Public Health, School of Medicine, Dentistry and Nursing, University of Glasgow, Glasgow, UK
[4]University Library, University of Glasgow, Glasgow, UK

**Correspondence to**
Eleanor R Dawson;
e.dawson@ucl.ac.uk

## ABSTRACT

**Introduction** Improving oral health and reducing oral health inequalities is an important global health priority. 'Upstream interventions' are a vital part of the collective effort to reduce oral disease burdens, however it is a rather nebulous term. Furthermore, there is little evidence on the effectiveness, impact and sustainability of upstream interventions that have focused on oral health and wider public health measures that impact on oral health. The aim of this scoping review is to systematically map and synthesise evidence on the effectiveness, impact and sustainability of upstream interventions on population oral health and reducing socioeconomic oral health inequalities.

**Methods and analysis** This scoping review will be conducted in accordance with the Joanna Briggs Institute methodology and the Preferred Reporting Items for Systematic Reviews and Meta-Analyses Extension for Scoping Reviews checklist. A detailed search strategy will be used to conduct a comprehensive search of electronic databases: Scopus, Embase and MEDLINE, PsycINFO and CINAHL, ASSIA and Cochrane Database of Systematic Reviews. A search of grey literature will also be completed to identify relevant dissertations, governmental reports and evaluations of implemented policies. Identification and extraction of data will be performed by two pairs of reviewers. Oversight and feedback will be provided by an independent expert advisory group.

**Ethics and dissemination** This study will review published and available grey literature and does not require an ethics review. The scoping review protocol has been registered with the Open Science Framework. The final report will be circulated and disseminated through publication and feed into the work of the ongoing Lancet Commission on Oral Health. Due to the policy relevance of this work, discussions will take place with key stakeholders regarding the implications of the findings for future policy development.

## STRENGTHS AND LIMITATIONS OF THIS STUDY

⇒ This scoping review will adopt a robust methodology in accordance with standard guidelines and checklists.
⇒ The review aims to use seven electronic databases and a tailored search strategy to retrieve as many relevant published works as possible.
⇒ The review will include literature published in any language and in the grey literature to create a comprehensive picture of upstream interventions.
⇒ The review will assess the broad evidence base for upstream interventions on oral health outcomes and socioeconomic inequalities, but will not undertake a detailed evaluation of interventions.

## INTRODUCTION

Oral diseases are a highly prevalent global public health problem, affecting approximately 3.5 billion people, which equates to almost half of the world's population.[1] Oral diseases (such as dental caries, periodontal disease and oral cancer) are largely preventable and in recent decades some improvements in oral health have been observed predominantly in high-income countries.[2] However, dental caries in many low-income and middle-income countries appears to be on the increase, linked to economic development and increasing availability and consumption of free sugars.[3] Even in high-income countries with lower disease burdens, dental services are largely treatment oriented without fully addressing the underlying causes of oral diseases.[4]

Poor oral health may lead to profound impacts on individuals and wider society. Pain or sepsis,[4] lower self-esteem,[5] reduced school attendance and poorer educational performance,[6] poorer quality of life[7] and reduced work productivity[8] are all common impacts of oral diseases. There are also significant economic costs associated with dental treatment and lost productivity, amounting to approximately US$544 billion globally in 2015.[9] Oral diseases disproportionately affect poorer, socially disadvantaged and marginalised groups; such differences are deemed to be avoidable, unfair and unjust in modern



society.[3] Furthermore, stark social gradients exist between socioeconomic position and the prevalence and severity of oral conditions,[10 11] reflecting similar patterns of inequality seen in general health.

Contemporary concepts regarding the determinants of health now acknowledge the underlying influence of structural and societal factors in socially patterning individual health behaviours and ultimately in determining disease levels.[12] Indeed, McKinlay[13] originally used the metaphor of people being pushed into a flowing river to describe the causes, or upstream 'manufacturers' of ill health in comparison to the downstream causes of illness. However, this metaphor can also be employed to highlight the different intervention opportunities (upstream, midstream or downstream) for public health programmes and policies to improve population health.[14]

Downstream interventions (individual/clinical/behavioural actions) have often been implemented to combat oral diseases and have failed to significantly reduce oral health inequalities or achieve sustainable improvements in oral health outcomes.[15] For example, individual oral health education interventions tend to result in limited short-term positive changes in oral health-related behaviours and health literacy, with marginal long-term improvements in clinical outcomes.[16–18] Moreover, evidence has demonstrated that downstream interventions may adversely result in widening health inequalities between socioeconomic groups, as they tend to provide more benefit for groups who are already advantaged.[19] This widening of the health inequality gap is often referred to as the 'inverse prevention law'[20] and has been observed with oral health education interventions.[21 22]

Upstream interventions are those that seek to address the underlying causative factors that lead to poor health, typically through action on broader structural, political, economic and environmental determinants. The term 'upstream intervention' is often a nebulous phrase to define in practice, as there may be difficulty in identifying broader public health approaches as purely 'upstream', as well as complexity in accurately defining population coverage and effectiveness of interventions. Overall, upstream interventions attempt to achieve sustained health equity in a population, often outside of the sphere of healthcare systems.[23] Upstream interventions generally aim to tackle broader social determinants (such as education, housing, access to safe drinking water and healthy food), or may create disincentives to engage in harmful health-related behaviours, such as fiscal policies to reduce consumption of sugary drinks.[24] Upstream interventions are generally initiated by national governments or policy makers and operate in a top-down manner across whole populations.

A broad consensus now supports the need for a combination of downstream, midstream and upstream interventions in order to effectively prevent oral diseases and promote oral health equity across the population.[4 25] However, questions remain over the extent of evidence surrounding the effectiveness of upstream interventions in promoting oral health and reducing socioeconomic health inequalities. This review will include the following upstream interventions for consideration: the implementation of water fluoridation schemes, regulations and fiscal measures aimed at reducing the consumption of tobacco, alcohol and sugar and interventions aimed at enhancing population welfare through improvement of housing, education and healthcare access. These upstream policies are those most commonly identified in the literature, however, the identification of other upstream interventions during the conduct of this review will also be included.

Although there are a wide range of persistent inequalities in oral health (such as inequalities between and within countries, ethnic differences and poorer outcomes in various vulnerable groups), this scoping review will focus on the impact of upstream interventions on promoting oral health and reducing socioeconomic oral health inequalities.

## METHODS
A scoping review has been chosen as the most appropriate form of evidence synthesis for this area of research as it will incorporate a broader scope and less restrictive inclusion criteria than a systematic review.[26] The proposed work of this review will be linked to the ongoing Lancet Commission on Oral Health, which includes an oral health inequalities workstream, and the findings will be shared with other key stakeholders regarding the possible implications for policy development.

This scoping review will be conducted in line with the Joanna Briggs Institute (JBI) methodology[26] and the Preferred Reporting Items for Systematic Reviews and Meta-Analyses Extension for Scoping Reviews (PRISMA-ScR) checklist.[27]

A preliminary search of Scopus, Google Scholar, Cochrane Database of Systematic Reviews and *JBI Evidence Synthesis* was conducted in June 2021. No ongoing or previously published scoping reviews on this topic were identified. Oversight and feedback of the methodology will be provided by an independent expert advisory group.

This scoping review study commenced in May 2021 with a planned completion date of July 2022.

### Scoping review questions
The aim of this scoping review is to identify the current literature that documents implemented upstream interventions and their effect on promoting oral health and reducing socioeconomic oral health inequalities. It will aim to address the following questions:
► What upstream interventions have been implemented to specifically target a reduction in socioeconomic oral health inequalities and promote oral health?
► What relevant and related public health upstream interventions, for example, tobacco control policies have been implemented to reduce socioeconomic health inequalities and promote health?

► Which upstream interventions have had an impact, are effective and are sustainable in reducing socioeconomic oral health inequalities and improving population oral health?

## Search strategy

The search strategy will aim to retrieve both published studies and relevant grey literature. An initial limited search of Scopus and PubMed was undertaken to identify articles relevant to the topic. The text words contained in the titles and abstracts of relevant articles and the index terms used to describe the articles will be used to develop a full search strategy on: Scopus, Embase and MEDLINE (via Ovid), PsycINFO and CINAHL (via EBSCOhost), ASSIA (via ProQuest) and Cochrane Database of Systematic Reviews (via the Cochrane Library).

Sources of grey literature to be searched include National Technical Information Service (NTIS), National Institute for Health and Care Excellence Evidence Search, TRIP, EThOS and WorldCat. The websites of relevant healthcare organisations will also be searched, for example, WHO, Public Health England, World Bank and The Health Foundation. Relevant national-level or state-level government policy documents will be examined. A manual search of bibliographies of selected articles will be undertaken by the research team to extract any relevant literature or identify grey literature.

Given the difficulty defining upstream interventions, a multistranded search method[28] will be employed, with each strand representing an intervention of interest, namely: health inequalities, health disparities, health inequities, oral health promotion, tobacco control, sugar tax, upstream action, food advertising, food labelling and public health (online supplemental appendix A). The dental and oral health elements of the search will be adapted from the strategies employed by Waldron *et al*[29] and Arora *et al*.[30] The sugar tax strand of the search will use the strategies of Pfinder *et al*[31] and the equity-focused strand will use the strategies of Prady *et al*.[32]

## Study selection
### Inclusion criteria
#### Participants and context
Only studies that include population-wide polices or studies that target certain population groups will be included in this review. This includes populations that have been specifically targeted for the intervention or the general population. There is no minimum or maximum number for the size of the population group or age restrictions. The review will be limited to literature that includes implemented upstream interventions. The review will include all global settings where upstream interventions have been brought in to directly impact population oral health.

#### Concepts and definitions
The primary concept in this review is 'upstream interventions'. There is no unified definition of this term, but this scoping review has used a definition collating input from a variety of previous publications.[13–15 25] Upstream interventions will be defined as 'strategies or policies that seek to address structural or environmental factors that impact population oral health and socioeconomic oral health inequalities. This will include population-level approaches, such as fiscal measures, regulation and legislation'.

Oral health inequalities are unequal and unfair differences in oral health outcomes across different groups in society.[4] Such differences often disproportionally affect poorer and socially disadvantaged members of society. The association between poor oral health and socioeconomic status is well documented and is prevalent throughout the life course and across populations regardless of national income status[4]

#### Sources and types of studies
This scoping review will include peer-reviewed published journal articles, and grey literature including relevant government reports and policy documents. There is no limit on the date of publication. This study will include publications that have not been published in English and a complete translation will be sought. However, if it is not possible to obtain a full English translation the paper will be excluded and its omittance recorded in the final report. This review will consider a variety of study designs, including experimental, quasi-experimental, before and after studies and observational studies (including cohort studies, case-control studies and cross-sectional studies). In addition, systematic reviews will be included in this review, providing they meet the inclusion criteria. The review will also include other relevant public health policies and interventions, for example, sugar and tobacco control measures.

#### Exclusion criteria
Studies reporting on clinical interventions delivered in clinical settings and restricted to specific patient populations will be excluded from this review. Studies only reporting on oral health knowledge and attitudes and reviews of websites or industry documents will also not be included.

## Patient and public involvement
This scoping review protocol was written with no patient or public involvement.

## Data extraction
The final data from the search strategy will be deposited into the database Rayyan (http://rayyan.qcri.org), where two pairs of reviewers will independently screen the titles and abstracts of each citation using the inclusion and exclusion criteria to identify eligible articles for full review. Full texts will be extracted into a data extraction form developed by the research team in Microsoft Word. The data extracted will include the following details: study characteristics (first author, publication year, country, published language, population size and

characteristics, duration and setting), type of upstream intervention, outcomes and key findings relevant to the review question. The data extraction tool will be modified and revised as necessary during the process of extracting data. If appropriate, authors will be contacted to request missing or additional data, where required.

## Collating, summarising and reporting the data

All of the relevant screened articles will be stored in the reference manager, Zotero (2020/V.5.0.88). The summarised characteristics of the extracted studies will be presented visually in tabular form due to the large number of studies that will be found. Results from the selected studies will also be presented in a narrative format which will relate directly to the review questions and aims.

## ETHICS AND DISSEMINATION

The final protocol has been submitted for registration with Open Science Framework. A detailed final report will be produced on the review methods, results and recommendations following completion of the scoping review. A final paper presenting the review results will be also prepared for publication in a peer-reviewed journal on completion of the review. The findings from this review will feed directly into the Lancet Commission on Oral Health and will be highlighted in the Commission Report depending on the final structure of the report. Due to the strong policy relevance of this work, discussions will take place with the funder (The Borrow Foundation) on completion of the review to decide whether to hold a dissemination workshop, as recommended by Levac *et al*[33] with key policy stakeholders regarding the relevance of the findings for future policy development. The findings from this review may also be presented through other relevant forums, such as international conferences or scientific meetings.

**Acknowledgements** The research team would like to thank the expert advisory group for their input into the development of the protocol and the scoping review design and methodology. The advisory group is composed of Professor Roger Keller Celeste (UFRGS, Brazil), Dr Manu Raj Mathur (University of Liverpool, UK/ Public Health Foundation of India), Dr Mary Jane McCallum (Canada), Professor Robert J. Weyant (University of Pittsburgh, USA), Dr Jenny Godson and Dr Semina Makhani (Public Health England). The research team would also like to thank Sana Daniyal and Afshan Mirza for their support in undertaking the initial screening and reviewing of eligible papers.

**Contributors** RW, MS, BD and LM conceived the idea for the review. RW, ERD, MS, BD, PC and LM all contributed to the development of the protocol. ERD, RW and MS drafted the original manuscript. RW, LM, PC and BD edited and revised the manuscript. ERD conducted the preliminary search and PC provided expertise on the final search strategy. All authors edited and approved the final text prior to submission for publication.

**Funding** This work was supported by The Borrow Foundation (UK Charity No: 1060308). The Borrow Foundation has no further role in the conduct of the review other than providing the designated funding.

**Competing interests** None declared.

**Patient and public involvement** Patients and/or the public were not involved in the design, or conduct, or reporting, or dissemination plans of this research.

**Patient consent for publication** Not applicable.

**Ethics approval** Not applicable.

**Provenance and peer review** Not commissioned; externally peer reviewed.

**Data availability statement** Data are available in a public, open access repository. Data are available at: https://doi.org/10.17605/OSF.IO/5ZY9G.

**ORCID iDs**
Eleanor R Dawson http://orcid.org/0000-0002-7842-583X
Michelle Stennett http://orcid.org/0000-0003-1731-9854
Blánaid Daly http://orcid.org/0000-0001-7748-6940
Lorna M D Macpherson http://orcid.org/0000-0002-7227-2991
Paul Cannon http://orcid.org/0000-0001-8721-1481
Richard G Watt http://orcid.org/0000-0001-6229-8584

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
