## [Reviewer comments · BMJ Open]

ARTICLE DETAILS

TITLE (PROVISIONAL)	Upstream Interventions to Promote Oral Health and Reduce Socio-economic Oral Health Inequalities: A Scoping Review Protocol
AUTHORS	Dawson, Eleanor; Stennett, Michelle; Daly, Blánaid; Macpherson, Lorna; Cannon, Paul; Watt, Richard

VERSION 1 – REVIEW

REVIEWER	Z Marshman University of Sheffield Faculty of Medicine Dentistry and Health
REVIEW RETURNED	23-Dec-2021

GENERAL COMMENTS	This manuscript describes a protocol for a scoping review of upstream intervention to improve oral health and reduce inequalities. The paper itself provides a very well written and useful overview of this topic in the Introduction section and a thorough description of the proposed method. The authors should consider the following minor suggestions: Abstract: In line 1, page 3 the authors state “upstream interventions that have previously targeted oral health” – the authors may wish to rephrase the sentence to avoid confusion about the use of the word targeted – as it can be used to distinguish between universal and targeted interventions – perhaps focused on oral health may be clearer? In line 30, page 3 in the Ethics and dissemination section, the authors state the study will only use published literature when some of the grey literature may not be published. Introduction In the sentence of the paragraph starting line 23, page 4 the authors should consider emphasising they are referring to improvements in oral health at the population level. The section providing the justification for the use of a scoping review method on page 5 could be moved to the Method section. Method The definition of upstream interventions is not needed in both the Introduction and the Method section. The authors should reference the method of scoping review they are undertaking. This is particularly relevant as some methods (proposed by Levac et al) include a consultation phase in which the planned stakeholder engagement could be carried out.
---

REVIEWER	Mark Keboa McGill University, Oral Health and Society
REVIEW RETURNED	23-Jan-2022

GENERAL COMMENTS	Dear Author, Thank you for the well-written scoping review protocol on interventions to promote oral health and reduce oral health inequalities. After a careful review of the manuscript, I have only one issue that requires clarification: • How do you plan to manage publications in languages other than English? Thank you.
---

VERSION 1 – AUTHOR RESPONSE

Reviewer: 1

Dr. Z Marshman, University of Sheffield Faculty of Medicine Dentistry and Health

Comments to the Author:

This manuscript describes a protocol for a scoping review of upstream intervention to improve oral health and reduce inequalities. The paper itself provides a very well written and useful overview of this topic in the Introduction section and a thorough description of the proposed method. The authors should consider the following minor suggestions:

Abstract:

In line 1, page 3 the authors state “upstream interventions that have previously targeted oral health” – the authors may wish to rephrase the sentence to avoid confusion about the use of the word targeted – as it can be used to distinguish between universal and targeted interventions – perhaps focused on oral health may be clearer?

Thank you for this comment. The use of the word ‘targeted’ has been replaced with ‘focused’ to provide more clarity in this sentence. – see page 2, lines 5-6 (marked document).

In line 30, page 3 in the Ethics and dissemination section, the authors state the study will only use published literature when some of the grey literature may not be published.

This is a valid point and we have now modified this statement to account for the inclusion of grey literature – see page 2, lines 20-21, (marked document).

Introduction

In the sentence of the paragraph starting line 23, page 4 the authors should consider emphasising they are referring to improvements in oral health at the population level.

We have now edited this sentence to highlight the population level – see page 4, line 14, (marked document).

The section providing the justification for the use of a scoping review method on page 5 could be moved to the Method section.

We have now moved the text justifying the use of a scoping review to the Method section – see page 5, lines 2-7 (marked document).

Method

The definition of upstream interventions is not needed in both the Introduction and the Method section.

Thanks for highlighting the repetition of text defining upstream interventions. We have now deleted the definition sentence from the Introduction – see page 4, lines 3-6 (marked document).

The authors should reference the method of scoping review they are undertaking. This is particularly relevant as some methods (proposed by Levac et al) include a consultation phase in which the planned stakeholder engagement could be carried out.

Thank you for this comment. We have reiterated that this scoping review follows the Joanna Briggs Institute Methodology and PRISMA ScR Checklist in the methods section and included relevant references. We have included a reference to Levac et al (2010) in reference to the workshop we aim to hold as part of our dissemination work - see page 8, line 1 (marked document).

Reviewer: 2

Dr. Mark Keboa, McGill University

Comments to the Author:

Dear Author,

Thank you for the well-written scoping review protocol on interventions to promote oral health and reduce oral health inequalities. After a careful review of the manuscript, I have only one issue that requires clarification:

- How do you plan to manage publications in languages other than English?

Thank you for this comment which does require clarification. This scoping review will include titles and abstracts that have been published in English for screening. If the full article is not published in English but is suitable for inclusion, a full translation will be sought. If this is not possible, the article will be excluded due to failure to obtain an English translation. - see page 6, line 37-40 (marked document).

VERSION 2 – REVIEW

REVIEWER	Z Marshman University of Sheffield Faculty of Medicine Dentistry and Health
REVIEW RETURNED	04-May-2022
GENERAL COMMENTS	The authors have adequately addressed my previous comments.